# Association of Androgen Deprivation Therapy with Osteoporotic Fracture in Patients with Prostate Cancer with Low Tumor Burden Using a Retrospective Population-Based Propensity-Score-Matched Cohort

**DOI:** 10.3390/cancers15102822

**Published:** 2023-05-18

**Authors:** Sung Han Kim, Ye Jhin Jeon, Jean Kyung Bak, Bit-Na Yoo, Jung-Wee Park, Yong-Chan Ha, Young-Kyun Lee

**Affiliations:** 1Department of Urology, Urologic Cancer Center, Research Institute and Hospital of National Cancer Center, Goyang 10408, Republic of Korea; snuh19@naver.com; 2Department Statistics, Yonsei University, Seoul 03722, Republic of Korea; 3National Evidence-Based Healthcare Collaborating Agency (NECA), Seoul 04933, Republic of Korea; 4Department of Orthopaedic Surgery, Seoul National University College of Medicine, Seoul National University Bundang Hospital, Seognam 13620, Republic of Korea; 5Department of Orthopaedic Surgery, Seoul Bumin Hospital, Seoul 07590, Republic of Korea

**Keywords:** prostate cancer, fracture, osteoporosis, androgen deprivation

## Abstract

**Simple Summary:**

This propensity-score-matched study showed significantly higher incidence and effects of osteoporotic fracture in patients with localized or regional SEER-staged prostate cancer treated with androgen deprivation therapy.

**Abstract:**

This study evaluated the effect of androgen deprivation therapy (ADT) on osteoporotic fractures (OF) and its prognostic effect on overall survival in patients with localized or regional prostate cancer (PC) using the Korean National Insurance Dataset. A total of 8883 pairs of 1:1 propensity-score-matched patients with localized or regional PC were retrospectively enrolled between 2007 and 2016. All patients underwent at least 1 year of follow-up to evaluate therapeutic outcomes. Multivariate analysis was performed to determine the prognostic effect of ADT on OF. During a mean follow-up of 47.7 months, 977 (3.43%) patients developed OF, and the incidences of hip, spine, and wrist fractures were significantly different between ADT and non-ADT groups (*p* < 0.05). The ADT group had a significantly higher incidence of OF (hazard ratio 2.055, 95% confidence interval 1.747–2.417) than the non-ADT group (*p* < 0.05), and the incidence of spine/hip/wrist OF was significantly higher in the ADT group regardless of the PC stage (*p* < 0.05). Multivariate analysis failed to show any significant difference in overall survival between the two groups (*p* > 0.05). ADT resulted in a significantly higher incidence of OF among patients with localized and regional PC, but the overall survival did not differ between ADT and non-ADT groups.

## 1. Introduction

Prostate cancer (PC) is the most frequent type of cancer in men and the second leading cause of cancer-related deaths after lung cancer in western countries, including in the United States, which had an annual incidence of 248,530 new PC cases and 34,130 estimated PC-related deaths in 2021 [1,2]. The standardized therapy for PC is systemic androgen deprivation therapy (ADT), which consists of either the removal of the primary prostate or the suppression of androgen-producing organs and the blockage of androgen release to achieve a castration level of <50 ng/dL of serum testosterone [2,3]. Long-acting luteinizing-hormone-releasing hormone agonists and gonadotropin-releasing hormone antagonists are currently the main types of ADT used in clinical practice with concomitant administration of anti-androgen to reduce the incidence of the clinical effects related to testosterone surge. More than 50% of patients with PC receive ADT as a mainstay therapy from the initial diagnosis until a curable state is achieved, and a 10-year follow-up is required to improve prognosis [3]. A longer duration of ADT in combination with other therapies for PC results in better survival outcomes.

However, like other therapies, long-term ADT results in adverse events that worsen the quality of life, as well as the survival outcomes, even after the cancer has been well controlled [4]. For example, hypogonadism due to androgen suppression negatively affects a variety of metabolic processes and organs, including the heart, vessels, bones, and muscles, resulting in cardiovascular disease, osteoporosis, sarcopenia, and bone fracture [5]. The significant negative relation between testosterone and cardiovascular disease has been shown, and age >5 years and body mass index (BMI) > 24 kg/m^2^ were risk factors for cardiovascular disease. Moreover, 3-year ADT has been reported to reduce bone mineral density by approximately 5–11% [6], and the risk of fracture increases proportionally to the duration of ADT [7]. Furthermore, the elevated risk of fracture with ADT has been found to be cumulatively associated with various disease conditions, ADT types, duration of dose levels, and fracture sites [7]. Several PC guidelines have recommended periodic bone marrow densitometry with adequate nutritional support for patients with a high risk of fracture and aged 50–69 years throughout ADT [8,9].

Recently, global statistics have shown differences in osteoporosis and its fracture rates between western and Asian patients, and the adverse effects of ADT on bone health in Asian patients has been the focus of attention owing to the rapidly increasing prevalence of PC in Asian countries. Different clinical practices and genetic ethnicities, a higher incidence of osteoporosis and osteoporotic fracture (OF), and more advanced and aggressive PC pathology were observed in Asian men compared with their Caucasian and European counterparts [10,11]. Moreover, ADT among Asian patients with PC has been reported to account for less than 3.0–8.7% of all cases of OF [12,13], whereas western studies have reported an overall fracture rate of 12.6–19.4% [10,14].

Recently, ADT has been extended to all patients with PC, from localized to metastatic PC [3]. Thus, the incidence of ADT-related bone health problems, including osteoporosis and OF, which affect survival among Asian patients with PC, is expected to increase [10,14]. Many studies have reported a significant association between the overall incidence of bone health problems and ADT among Asian patients with PC using the National Cancer Cohort dataset [13]. However, most studies were published when ADT was indicated only for patients with advanced and metastatic PC treated with or without radiation therapy, and the duration of ADT was short due to the short-term survival period of such patients (median 2 years). Recent therapeutic and preventive guidelines have extended the indication of ADT to localized PC, including those undergoing active surveillance. As a result, many patients receive long-term ADT, necessitating prophylaxis for OF [3,15]. Furthermore, only a few studies have considered the variability of the extent of the disease burden according to clinical stage and tumor burdens, and most papers evaluated the relations of ADT and OF in advanced and metastatic PC with high tumor burden [6,16].

Therefore, it is important to investigate the relationship between OF and ADT in PC patients with low tumor burden stratified by the cancer stage. This study applied the Surveillance, Epidemiology, and End Results (SEER) staging system to define the localized- and regional-stage PC cohort and compared the patients with or without ADT. This study aimed to evaluate the incidence and effects of OF and analyze the significant risk factors of OF and overall survival after adjusting the baseline differences using propensity score matching (PSM) in localized or regional SEER-staged Korean PC patients with and without ADT.

## 2. Materials and Methods

### 2.1. Ethical Statement

This retrospective, population-based study was exempted from Institutional Review Board (IRB) approval because it did not involve human subjects (Seoul National University of Bundang Hospital IRB No. X-1801-447-908). Moreover, the IRB waived the need for written patient consent because all data were anonymized and provided by the Korean National Insurance Database, which is managed by the Korean National Insurance Association. All study protocols were performed in accordance with the tenets of the ethical guidelines and regulations of the World Medical Association Declaration of Helsinki Ethical Principles for Medical Research Involving Human Subjects.

### 2.2. Population-Based Cancer Registry Database

The analytical methodology for anonymizing the Korean National Health Insurance System of Statistics (NHIS) database has been described in detail in our previous epidemiological studies [17,18]. Based on the International Classification of Diseases, Tenth Edition (ICD-10) diagnostic code C61, the NHIS provided data for 86,230 patients with PC aged >40 years who were diagnosed between 2007 and 2016. Previously diagnosed patients with multiple diagnosis dates were excluded, and only accurately classified existing and new patients were included. In addition, patients without complete information, including survival outcomes until 2018, were excluded, resulting in a final number of 53,528 (62.1%) patients with available survival outcomes according to the NHIS data (Figure 1).

### 2.3. Inclusion and Exclusion Criteria of Patient Selection

Among the 53,528 patients with PC, 28,490 (53.2%) were finally enrolled for the analysis after applying the following exclusion criteria: another cancer diagnosis during the follow-up period through the year 2018 (*n* = 2283), previous history of rheumatoid arthritis and steroid use (*n* = 6394), death within 6 months after PC diagnosis (*n* = 299), previous history of OF (*n* = 1138), history of distant/unknown SEER stage (*n* = 8020), history of more than two combined exclusion criteria among the above five criteria (*n* = 5250), and prescribed ADT for <90 days (*n* = 1654). The remaining 28,490 patients were classified into ADT (*n* = 11,696; 41.0%) and non-ADT (*n* = 16,794; 59.0%) groups.

### 2.4. Primary Outcome and Other Clinicopathological Variables

This study focused on ADT-related OF in the spine, hip, wrist, and shoulder as skeletal-related events. OF was identified by combining ICD codes for fractures, procedure codes, and radiographic codes, as previously described [17,18]. The ICD-10 codes used for spine fractures were M48.4, M48.5, M49.5, M80.88, S22.0, S22.1, S32.0, S32.7, and T08: those for hip fractures were S72.0 and S72; those for humerus fractures were S422 and S423; and those for wrist fractures were S525 and S526. Data on OF that occurred at least 6 months after ADT were collected.

Other clinicopathological parameters including underlying disease, BMI, Charlson comorbidity index (CCI), and medication use were selected according to the methods used in our previous epidemiological studies. Only the localized and regional SEER-staged classifications were selected to rule out distant and unknown SEER-staged PC. The primary therapeutic strategies for PC treatment included surgery and radiation, which were coded using the EDI-CD. We also added a no treatment group (active surveillance group), which had no history of any treatment except surgery, radiation, and chemotherapy.

During the study period, 84 hormonal agents approved by the Korean Food and Drug Administration, including gonadotropin-releasing hormone agents (goserelin, leuprorelin, and triptorelin acetate) and anti-androgen agents (flutamide and bicalutamide), were used for ADT. The final ADT group comprised patients treated with ADT for at least 6 months.

### 2.5. PSM Analysis

We performed 1:1 PSM between the ADT and non-ADT groups among a total of 17,766 patients or 8883 pairs. The mean differences between the ADT and non-ADT groups before and after PSM were 0.1485 and 0.001, respectively (Appendix A). The matched variables included age at cancer diagnosis, year of diagnosis, urbanicity (urban vs. rural), insurance type (self-employed insured, employee insured, medical aid beneficiary), income level, SEER stage (localized, regional), modified CCI, and primary therapy modalities (radiotherapy [RT], surgery), which were significantly different at baseline between the ADT and non-ADT groups. After matching, homogeneity between the groups was evaluated using the chi-squared test or McNemar’s test for categorical data, a paired t-test for continuous data, and analysis of variance to explain the effect of each confounding variable on the incidence of dementia.

To adjust for confounding variables, we performed regression analysis to test the differences in the resultant variables. Multivariate Cox analysis was used to define the effect of ADT and its duration on the incidence of OF during the study period. From the starting date of ADT, hazard ratios (HRs) and 95% confidence intervals (CI) for the occurrence of OF and death among the ADT groups were analyzed using Cox proportional hazard models adjusted for ADT, age at cancer diagnosis, year of diagnosis, and urbanicity (urban vs. rural). The insurance type (self-employed insured, employee insured, medical aid beneficiary), income level, SEER stage (localized, regional), CCI, and primary therapy modalities (radiotherapy, surgery, active surveillance) were used. *p*-values < 0.05 indicated statistical significance. All statistical analyses were performed using SAS (version 9.4, SAS Institute Inc., Cary, NC, USA).

## 3. Results

### 3.1. Demographics of Overall Population and Propensity-Score-Matched Patients

Among the 8883 matched pairs of localized and regional SEER-staged PC cohorts, the overall mean age and follow-up period were 68.64 (SD 6.79) years and 47.71 (±SD 29.90) months (Table 1, Appendix A). Moreover, 11,560 and 2963 patients had localized and regional PC, respectively, and 15.29%, 34.56%, 26.57%, and 23.58% of patients received RT, surgery, no treatment, and hormone therapy, respectively, as the primary treatment for PC. Additionally, 9.68% and 74.78% of patients with PC had modified CCI Grades 3 and 4, respectively. During the study period, 113 (16.5%) deaths occurred, and 977 (3.4%) patients experienced 1046 OFs, including 232 (22.2%) hip, 469 (44.8%) spine, 319 (30.5%) wrist, and 26 (2.5%) humeral lesions. Finally, 69 patients had a history of double fractures.

Among the baseline variables, even after PSM, statistically significant differences were observed in the year at diagnosis, urbanicity, insurance type, income level, SEER stage, CCI, primary therapy modality, OF, and follow-up duration between the two groups (*p* < 0.05), whereas age, death, and underlying diseases were not significantly different (*p* > 0.05) (Table 1, Appendix A). However, the mean group difference decreased from 0.1485 in 28,471 patients before PSM to 0.0001 in 17,766 patients after PSM (Appendix A).

### 3.2. Incidence of OF between the Two Groups

The ADT group had a significantly higher incidence of OF (5.23%) than the non-ADT group (2.48%) (*p* < 0.001), although the follow-up period was significantly longer in the non-ADT group (48.57 ± 30.45 months) than in the ADT group (46.38 ± 29.31 months) (*p* < 0.001, Table 1). The ADT group had a significantly higher incidence of OF than the non-ADT group (1340.461 vs. 611.945 per 100,000 person-years) at every fracture site (*p* < 0.001), except for humerus fractures (4100 vs. 4028, *p* = 0.328) (Table 2). The Cumulative incidence of OF also showed similar findings regarding the significant differences in the incidence of OF between the two groups in overall, hip, spine, and wrist lesions (*p* < 0.001, Figure 2).

### 3.3. Risk Factors of OF

In the multivariate analysis of risk factors of OF, ADT was a significant independent factor (HR 2.055; 95% CI 1.747–2.417) along with other significant factors such as age (HR 1.068; CI 1.054–1.082) and CCI Grade 4 (HR 2.038; CI 1.206–3.444) (*p* < 0.05, Table 3). ADT was also an independent risk factor for hip (HR 1.842; CI 1.298–2.615), spine (HR 2.080; CI 1.628–2.658), and wrist (HR 1.749; CI 1.316–2.324) fractures (*p* < 0.001) but not for humerus fractures (Appendix A). In addition, age, Grade 4 CCI, and history of bone densitometry increased the risk of hip and spine fractures (*p* < 0.05) but were insignificant for wrist fractures (*p* > 0.05).

The multivariate model adjusted by the significantly different baseline variables of each lesion showed that the ADT group had an approximately higher risk of OF in the hip, spine, and wrist than the non-ADT group, regardless of the localized and regional SEER stage (*p* < 0.05, Table 4). The highest risk of OF was observed for hip lesions, with an HR > 2.0 in the ADT group compared with that in the non-ADT group for localized (HR 2.286; CI 1.6–3.268) and regional PC (HR 2.284; CI 1.115–4.675), whereas the incidence of humeral fracture was not significantly different for either SEER stage between the two groups (*p* > 0.05).

### 3.4. Association of OF with Overall Survival and Death between the ADT and Non-ADT Groups

The cumulative overall survival rate analysis showed significant differences in all OF lesions except for humerus fractures between the ADT and non-ADT groups at, 1, 3, 5, and 7 years (*p* < 0.001, Appendix A). The overall survival rate remained >90%, even in patients with OF, until 7 years after the initial PC diagnosis.

The cumulative mortality curve among all patients, those with localized PC and those with regional PC, showed a significant difference in mortality rates between the ADT and non-ADT groups (Figure 3A). The non-ADT group had a significantly lower rate of mortality in patients with localized and regional PC (*p* < 0.001). The cumulative mortality curve of death showed insignificant differences between the OF and non-OF groups among patients with localized or regional PC (*p* > 0.05, Figure 3B). Finally, the cumulative mortality curve among the groups stratified by ADT and OF showed that the mortality rates differed significantly among all patients, those with localized PC and those with regional PC (*p* < 0.001, Figure 3C).

## 4. Discussion

Since the indications of ADT have been expanded to patients with localized and regional PC as a mainstay therapy for PC with or without primary cancer therapy, ADT-related bone health events have attracted attention from urologists [3]. Some Asian countries, such as Japan and Korea, have reported a higher rate of increase in PC among men [1,16,19]. In addition, Asians exhibit a significantly higher prevalence of osteoporosis than Westerners [14,19,20]. Therefore, OF likely impacts quality of life and survival outcomes proportionally to the duration of ADT [3,21]. However, a randomized controlled trial evaluating the effects of ADT on skeletal-related events would be difficult to perform due to ethical issues, including the lack of mainstay therapy for the non-ADT group. Therefore, a retrospective analysis with a large national cohort is a feasible way to evaluate the association of ADT with OF in a real-world setting.

This national, retrospective cohort study evaluated the impact of ADT on the incidence of OF, especially among patients with localized- and regional-stage PC. Many previous studies have evaluated mostly advanced and metastatic PC patients with a high tumor burden [22,23] or those with localized PC without any significantly subtyping clinical stages. However, this study selected patients with localized and regional SEER-staged PC to better analyze the data. The results showed a significantly higher overall incidence of OF in the ADT group (HR 2.055; 95% CI 2.747–2.417, *p* < 0.001) without any significant differences in overall survival during a follow-up period of nearly 5 years (47.71 ± SD 29.90 months) (*p* > 0.05, Table 3). In the 7-year follow-up analysis, the incidence of OF at the hip, spine, and wrist lesions were significantly affected by ADT (*p* < 0.05; Table 4). This implied the necessity of monitoring for OF in patients undergoing long-term ADT.

Clinicians often fail to apply routine preventive measures and administer prophylactic agents for OF in patients with localized or regional SEER-staged cancer undergoing ADT, especially in older patients. One retrospective study showed that only 40.3% of patients were prescribed osteoporosis medication within 1 year after hip fracture, and incidence of osteoporosis therapy decreased proportionally with age. [23] The reason for the lower rate of preventive measures for OF in PC patients is because older PC patients already take many different medications for other underlying diseases and are reluctant to take additional medication only for the prevention of OF, which has a low rate of occurrence. Additionally, clinicians did not encourage their patients to actively take preventive measures for OF because ADT-related OF seemed to have low incidence rates and did not affect cancer-specific survival during the PC therapy. Urologists encounter their patients with OF after their patients are successfully treated by orthopedic surgeons; this leads to urologists not encountering emergent OF cases.

There is no specific evidence based on clinical trials regarding the necessity of early intervention for OF in patients undergoing ADT [15,16], and the appropriate type of prophylaxis for OF has not been identified [9,24,25]. However, most PC patients are older, and recently improved survival due to diverse new hormonal agents, in combination with ADT, has increased the prevalence of PC worldwide. This necessitates educational programs and active recommendations of preventive agents by urologists. A retrospective PSM study from the SEER database including 54,953 patients with PC aged >66 years reported an OF rate of 17.5% after initiation of ADT, with a median fracture time of 31 months (15–56 months). The same study reported that dual energy X-ray absorptiometry (DEXA) screening rates were low among older men with localized or regional PC after initiation of ADT; however, bone mineral density scans showed a lower risk of OF [6]. Some of the recent recommendations for osteoporosis emphasize that fracture risk evaluation and interventions targeting risk factors of OF should be provided to all patients undergoing ADT [12,14]. Moreover, patients with a history of severe OF and/or a T-score < −2.5 should receive osteoporosis therapy such as bisphosphonate or denosumab.

However, no specific bone health guidelines for patients with localized or regional PC have been established. Early prophylactic intervention has not shown any beneficial role in clinical and economic aspects, and early monitoring of bone health might be feasible with the current modalities [6,25]. Additionally, patients with risk factors for OF such as advanced age (HR 1.068) and CCI Grade 4 (HR 2.038) (Table 3) should be identified for the administration of preventive measures. In addition, previous studies have found that radiation therapy as the primary therapy for localized or regional PC increased the rate of osteoporosis and OF when used in combination with ADT to almost twice the rate without ADT; the risk of OF increased proportionally with the duration of ADT [6,21,25]. A similar study by Shin et al. showed that primary treatment with ADT and surgery (HR 1.41), ADT and RT (HR 1.86), ADT only (HR 1.92), and metastatic tumors (HR 1.39) were significant risk factors of OF [26]. Another Korean population-based PSM study with a similar period of diagnosis and follow-up showed a significantly higher risk of OF (8.12%; HR 1.815; 95% CI 1.703–1.935) (*p* < 0.001) in the ADT group compared with the non-ADT group (7.08% and 5.04%, respectively) after age adjustment, and a longer duration of ADT was associated with an increased risk of osteoporosis and OF [22]. Similarly, a study with 24 months of DEXA follow-up for 50 patients with localized cancer and 55 with metastatic disease showed that a longer duration of ADT significantly increased the osteoporosis rate from 10% to 22% and decreased the normal bone density from 32% to 8% and 6%, respectively, resulting in OF [21,25].

Previous studies have identified several risk factors of OF among patients undergoing ADT, such as old age and high CCI [3,9,14,15,24]. Some guidelines have also suggested several risk factors of OF, such as previous history of frequent fracture and fall after 50 years of age, lower BMI, and current glucocorticoid therapy. Active risk assessment using DEXA or the Fracture Risk Assessment Tool (FRAX) (http://www.shef.ac.uk/FRAX, accessed on 15 May 2023) should be implemented [8]. The highest incidence of OF in this study occurred in the spine (HR 2.080; 44.8%), followed by the wrist (HR 1.842; 30.5%) and hip (HR 1.749; 22.2%). The spine and hip are major weight-bearing sites that are easily broken in patients with osteoporosis, while the wrist is the most vulnerable and fragile site near the hands when patients fall, as they are often used to relieve the impact of the fall [14]. In addition, the spine was found to be the most frequent site of bone disease progression when PC treatment failed [27]. Overall, these findings highlight the effect of ADT on OF and suggest that the spine, hip, and wrist should be monitored closely in high-risk patients with localized/regional PC with advanced age and CCI Grade 4 treated with ADT. For such patients, the use of multi-prophylactic agents should be considered according to the established clinical guidelines [9,15,24].

The insignificant effect of OF on survival might be easily explained by the cohort having a low tumor burden and lower risk of tumor recurrence (<30%), requiring no further secondary hormone targets, RT, or chemotherapy, which negatively affect bone metabolism as well as cancer cell genomics [6,28,29]. In the RTOG 0518 study, which evaluated the protective effect of zoledronic acid on high-risk patients and/or those with locally advanced PC treated with RT and ADT [28], a 5–8% decrease was observed in the lumbar spine, bilateral hip, and femoral neck in bone densitometry measurements by exposing larger areas, such as the proximal femur, in the radiation fields of pelvic RT, and an increased risk of hypogonadism-induced bone loss was observed from pelvic external beam RT [6]. In a study by van Oostwaard et al., despite a lower 10-year risk of OF, there was a high prevalence of vertebral fractures in men with PC undergoing ADT, and the importance of a systematic vertebral fracture assessment with BMD and FRAX to improve identification of true incident vertebral fractures during ADT was shown [30].

This study had several limitations. For example, this was a retrospective population study based on insurance claim data in the absence of specific laboratory and pathologic data, such as PSA, calcium, and phosphorus levels and Gleason score. Moreover, the follow-up time was inconsistent for each period of PC diagnosis, with only one year of follow-up for the recently diagnosed cohort. We also did not consider the protective effect of anti-resorptive and supplementary medications, such as vitamin D and calcium [8,9]. Moreover, the diagnostic modalities of osteoporosis, social information, such as smoking and drinking status, and the prognostic effect of disease progression were not considered because excluding patients without this data available in the insurance claim database would have resulted in excluding a significant proportion of the cohort, ultimately leading to a non-representative cohort and selection bias. Nevertheless, this study attempted to adjust for all the baseline differences by PSM after excluding all the potential variables with detrimental effects on bone health in localized- and regional-stage PC patients with/without ADT by SEER stage. This study showed the significant prognostic effect of ADT and age on OF, suggesting the necessity for a thorough evaluation plan for bone metabolism in old-age patients with localized/regional-stage PC with long-term ADT use, especially those with high comorbid underlying histories. The hip, spine and wrist can be appropriate lesions to follow for OF prevention.

## 5. Conclusions

This study showed a significant relationship between ADT and incidence of OF in localized and regional SEER-staged PC patients, even during a long-term follow-up. However, no significant differences were observed in the overall survival between ADT and non-ADT groups. The spine, hip, and wrist lesions were the most significant OF sites in patients undergoing ADT. In future, larger controlled studies may be needed to evaluate the direct effect of long-term ADT in PC patients with low tumor burden, such as early localized- and regional-stage PC.

## Figures and Tables

**Figure 1 cancers-15-02822-f001:**
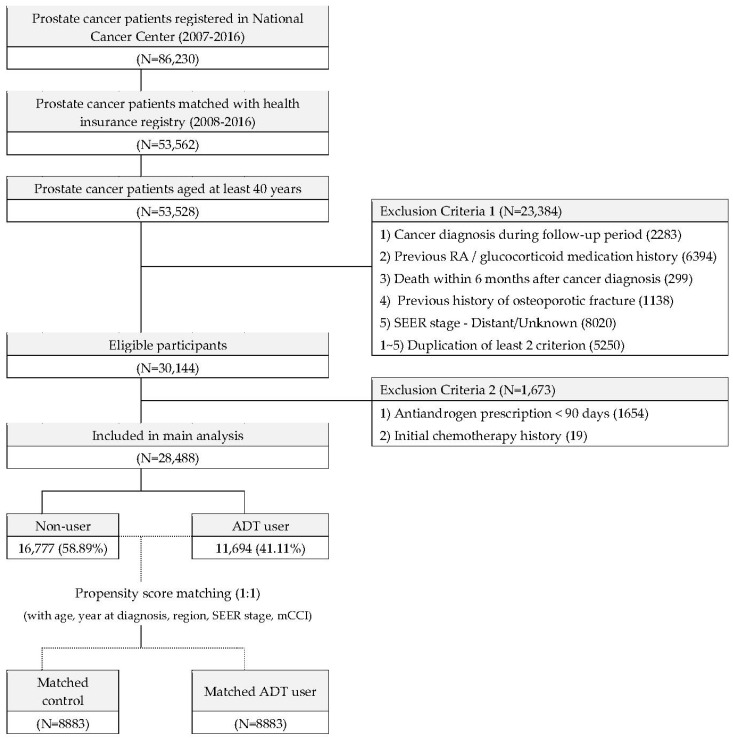
Flowchart of patient inclusion and exclusion.

**Figure 2 cancers-15-02822-f002:**
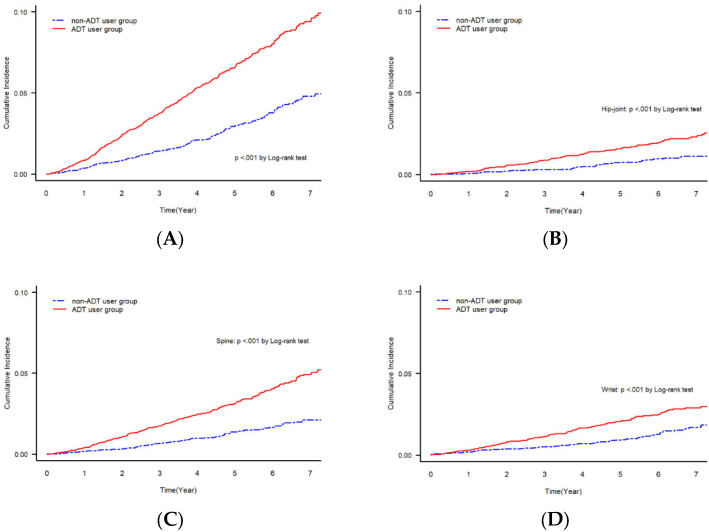
Cumulative incidence of OF with log-rank test between ADT and non-ADT users according to fracture sites: (**A**) overall incidence, (**B**) hip, (**C**) spine, (**D**) wrist, and (**E**) humerus.

**Figure 3 cancers-15-02822-f003:**
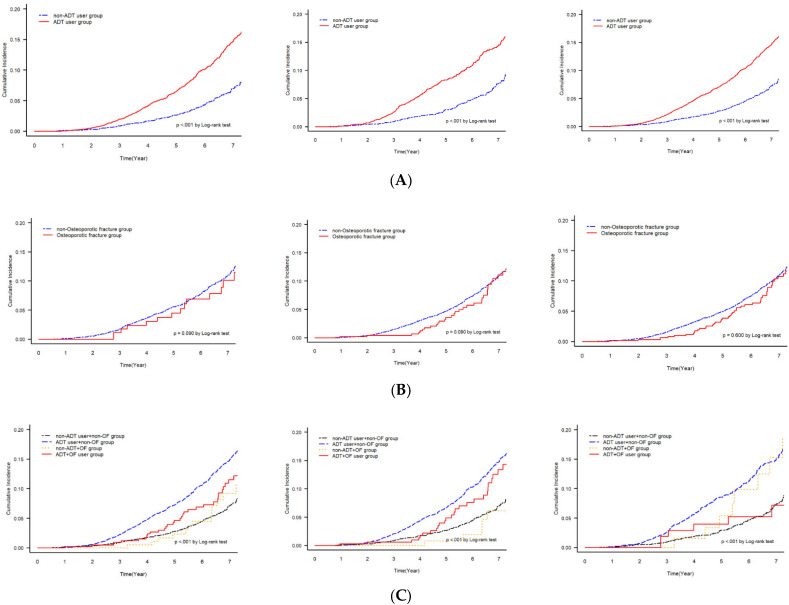
Cumulative mortality of overall patients, localized patients, and regional-stage patients with log-rank test according to (**A**) presence of ADT use, (**B**) osteoporotic fracture, and (**C**) osteoporotic fracture and ADT use.

**Table 1 cancers-15-02822-t001:** Baseline demographics of the propensity-score-matched cohort with prostate cancer (N = 17,766).

		Total		Non-ADT User		ADT User		*p*-Value
		(*n* = 17,766)		(*n* = 8883)		(*n* = 8883)		
Variables		N	(%)	N	(%)	N	(%)	
Age (years)								
	Mean (SD)	68.64	(6.79)	68.56	(6.59)	68.71	(6.99)	0.154
Year at diagnosis								
	2008~2010	5854	32.95%	2683	30.20%	3171	35.70%	<0.001
	2011~2013	5716	32.17%	3056	34.40%	2660	29.94%	
	2014~2016	6169	34.88%	3144	35.39%	3052	34.36%	
Urbanicity								
	Urban (Metropolitan areas)	12,284	69.14%	6231	70.15%	6053	68.14%	0.038
	Rural (Other areas)	5482	30.86%	2652	29.85%	2830	31.86%	
Insurance type								
	Self-employed insured	5359	30.16%	2773	31.22%	2586	29.11%	<0.001
	Employee insured	11,788	66.35%	5913	66.57%	5875	66.14%	
	Medical aid beneficiary	619	3.48%	197	2.22%	422	4.75%	
Income level (quintile)								
	upper > 80%	2079	11.70%	921	10.37%	1158	13.04%	<0.001
	61–80%	1711	9.63%	793	8.93%	918	10.33%	
	41–60%	2089	11.76%	1001	11.27%	1088	12.25%	
	21–40%	3480	19.59%	1700	19.14%	1780	20.04%	
	lower < 20%	7530	42.38%	4152	46.74%	3378	38.03%	
	Medicaid	877	4.94%	316	3.56%	561	6.32%	
modified CCI								
	0–1	1103	6.21%	630	7.09%	473	5.32%	<0.001
	2	1658	9.33%	814	9.16%	844	9.50%	
	3	1720	9.68%	941	10.59%	779	8.77%	
	4+	13,285	74.78%	6498	73.15%	6787	76.40%	
SEER summarized stage								
	Localized	11,560	65.07%	5640	63.49%	5920	66.64%	<0.001
	Regional	2963	34.93%	3243	36.51%	2963	33.36%	
Radiotherapy (Yes)		2717	15.29%	681	7.67%	2036	22.92%	<0.001
Surgery (Yes)		6140	34.56%	3660	41.20%	2480	27.92%	<0.001
Osteoporotic Fracture (Event)		685	3.86%	220	2.48%	465	5.23%	<0.001
Follow-up Period (MONTH)								
	Mean (SD)	47.71	29.90	48.57	30.45	46.86	29.31	<0.001

CCI—Charlson comorbidity index; SEER—Surveillance, Epidemiology, and End Results.

**Table 2 cancers-15-02822-t002:** Incidence rate of osteoporotic fracture in prostate cancer patients between the ADT and non-ADT groups.

Androgen Deprivation Therapy	Number of Events	Mean FU Period (Years)	Incidence Rate/100,000 Py	(95% CI)	*p*-Value
Osteoporotic fracture (Composite)					
Non-user (*n* = 8883)	220	4.047	611.945	(533.748–695.409)	<0.001
ADT user (*n* = 8883)	465	3.905	1340.461	(1221.378–1465.004)	
Osteoporotic fracture (Subtype: Hip)					
Non-user (*n* = 8883)	49	4.091	134.828	(99.746–175.11)	<0.001
ADT user (*n* = 8883)	114	4.008	320.186	(264.114–381.575)	
Osteoporotic fracture (Subtype: Spine)					
Non-user (*n* = 8883)	99	4.078	273.261	(222.093–329.654)	<0.001
ADT user (*n* = 8883)	233	3.972	660.424	(578.340–747.875)	
Osteoporotic fracture (Subtype: Wrist)					
Non-user (*n* = 8883)	80	4.082	220.642	(174.956–271.549)	<0.001
ADT user (*n* = 8883)	143	3.982	404.312	(340.762–473.215)	
Osteoporotic fracture (Subtype: Humerus)					
Non-user (*n* = 8883)	7	4.100	19.218	(7.727–35.857)	0.328
ADT user (*n* = 8883)	11	4.028	30.744	(15.348–51.400)	

Abbreviations: FU—follow-up; Py—person-year; CI—confidence interval; ADT—androgen deprivation therapy.

**Table 3 cancers-15-02822-t003:** Univariate and Multivariate Cox proportional hazard ratio model for risk of osteoporotic fracture.

Variables	Univariate Model			Multivariate Model		
	HR	(95% CI)	*p*-Value	HR	(95% CI)	*p*-Value
Androgen deprivation therapy						
Non-user	1.000	ref	<0.001	1.000	ref	<0.001
ADT user	2.214	1.885–2.599		2.055	1.747–2.417	
Age at index date	1.076	1.063–1.090	<0.001	1.068	1.054–1.082	<0.001
Year at diagnosis						
2008~2010	1.000	ref	-	1.000	ref	-
2011~2013	0.769	0.642–0.922	0.004	0.882	0.725–1.058	0.177
2014~2016	0.958	0.729–1.259	0.761	0.961	0.731–1.263	0.774
Urbanicity						
Urban (Metropolitan areas)	0.963	0.820–1.130	0.642	1.033	0.879–1.214	0.693
Rural (Other areas)	1.000	ref	-	1.000	ref	-
Income level (quintile)						
upper > 80%	1.000	ref	-	1.000	ref	-
61–80%	0.922	0.650–1.309	0.650	0.985	0.695–1.397	0.934
41–60%	1.238	0.911–1.682	0.172	1.286	0.946–1.746	0.108
21–40%	0.961	0.719–1.283	0.786	0.964	0.722–1.288	0.804
lower < 20%	0.945	0.730–1.222	0.665	0.917	0.708–1.187	0.510
Unknown or Medical aid	1.721	1.219–2.429	0.002	1.263	0.892–1.789	0.188
SEER stage						
Localized	1.000	ref	-	1.000	ref	-
Regional	0.769	0.651–0.908	0.002	1.021	0.858–1.215	0.8112
modified CCI						
0–1	1.000	ref	-	1.000	ref	-
2	1.035	0.538–1.992	0.9169	0.826	0.432–1.577	0.5621
3	1.754	0.963–3.196	0.066	1.497	0.827–2.707	0.183
4+	2.899	1.707–4.925	<0.001	2.038	1.206–3.444	0.008

ADT—androgen deprivation therapy; CCI—Charlson comorbidity index; SEER—Surveillance, Epidemiology, and End Results.

**Table 4 cancers-15-02822-t004:** Univariate and Multivariate Cox proportional hazard ratio model for risk of osteoporotic fracture according to SEER summarized stage.

Androgen Deprivation Therapy		Age-Adjusted Model			Fully Adjusted Model		
		HR	(95% CI)	*p*-Value	HR	(95% CI)	*p*-Value
Osteoporotic fracture (Composite)							
SEER stage: Localized	Non-user	1.000	ref	-	1.000	ref	-
	ADT user	2.239	1.844–2.719	<0.001	2.159	1.776–2.624	<0.001
SEER stage: Regional	Non-user	1.000	ref	-	1.000	ref	-
	ADT user	1.920	1.435–2.568	<0.001	1.816	1.347–2.448	<0.001
Osteoporotic fracture (Subtype: Hip)							
SEER stage: Localized	Non-user	1.000	ref	-	1.000	ref	-
	ADT user	2.483	1.744–3.534	<0.001	2.286	1.6–3.268	<0.001
SEER stage: Regional	Non-user	1.000	ref	-	1.000	ref	
	ADT user	2.267	1.12–4.59	0.0229	2.284	1.115–4.675	0.0239
Osteoporotic fracture (Subtype: Spine)							
SEER stage: Localized	Non-user	1.000	ref	-	1.000	ref	-
	ADT user	2.738	2.113–3.548	<0.001	2.534	1.949–3.295	<0.001
SEER stage: Regional	Non-user	1.000	ref	-	1.000	ref	-
	ADT user	1.657	1.132–2.425	0.0094	1.588	1.078–2.341	0.0193
Osteoporotic fracture (Subtype: Wrist)							
SEER stage: Localized	Non-user	1.000	ref	-	1.000	ref	-
	ADT user	2.072	1.553–2.765	<0.001	2.008	1.494–2.698	<0.001
SEER stage: Regional	Non-user	1.000	ref	-	1.000	ref	-
	ADT user	2.004	1.289–3.116	0.002	1.970	1.25–3.106	0.0014
Osteoporotic fracture (Subtype: Humerus)							
SEER stage: Localized	Non-user	1.000	ref	-	1.000	ref	-
	ADT user	1.519	0.517–4.459	0.4471	1.321	0.437–3.994	0.622
SEER stage: Regional	Non-user	1.000	ref	-	1.000	ref	-
	ADT user	1.762	0.451–6.887	0.6627	1.844	0.427–7.97	0.3756

## Data Availability

All data are available upon request after the review of the IRB of Bundang Seoul National University Hospital (eirb@snubh.or.kr) and the corresponding author (ykleemd@gmail.com).

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
