# Peer review of "Association of Androgen Deprivation Therapy with Osteoporotic Fracture in Patients with Prostate Cancer with Low Tumor Burden Using a Retrospective Population-Based Propensity-Score-Matched Cohort"

_cancers, 2023, doi:10.3390/cancers15102822_

Round 1

Reviewer 1 Report

The authors should be congratulated for the interesting topic discussed.

Despite the countless successes in the clinical field when it comes to approaching to PCa, we have not to forget comorbidities that drug therapy such as ADT, although correct, can add to the patient, worsening their quality of life.

Having the chance to administer a counter-therapy in these cases could be a great solution for both patient and specialist, even thou randomized controlled studies about it are still lacking.

I believe that the study has sufficient merit to be considered for publication, although major revisions are required.

1.    Methods and methodology are robust.

2.    Results, conclusions, and limitations are well presented.

3.    I would suggest to the authors should to give more detailed information on the side effects of antiandrogenic therapy, also emphasizing better those who are the cardiovascular ones. A lecture on this interesting study (https://doi.org/10.3389/fendo.2021.695170) could enhance the scientific value of the paper.

4.    The authors should provide more information about PCa epidemiology. This paper (https://doi.org/10.3390%2Fijms22189971) can meet these needs. A lecture is suggested.

Author Response

  • • Manuscript ID: cancers-2272182
  • Title: Association of androgen deprivation therapy with osteoporotic fracture

in patients with localized and regional prostate cancer using a retrospective

population-based propensity score-matched cohort

Reply to Reviewer #1’s comments

We thank reviewer #1 for their time spent reviewing our manuscript and valuable comments.

  1. Methods and methodology are robust.

Response: Thank you for the comment. We have revised the methods section further to add more detailed information regarding the data extraction protocol from the National Insurance Dataset.

  1. Results, conclusions, and limitations are well presented.

Response: Thank you for your encouraging comment.

  1. I would suggest to the authors should to give more detailed information on the side effects of antiandrogenic therapy, also emphasizing better those who are the cardiovascular ones. A lecture on this interesting study (https://doi.org/10.3389/fendo.2021.695170) could enhance the scientific value of the paper.

Response: Thank you for this comment. We have added some information regarding the side effects of ADT on cardiovascular disease in the introduction section and have included the corresponding reference (page 2 line 64-69 and page 9 lines 401)

Ref> 5.Sciarra A, Busetto GM, Salciccia S, Del Giudice F, Maggi M, Crocetto F, Ferro M, De Berardinis E, Scarpa RM, Porpiglia F, Carmignani L, Damiano R, Artibani W, Carrieri G. Does Exist a Differential Impact of Degarelix Versus LHRH Agonists on Cardiovascular Safety? Evidences From Randomized and Real-World Studies. Front Endocrinol (Lausanne). 2021 Jun 14;12:695170. doi.org/10.3389/fendo.2021.695170

  1. The authors should provide more information about PCa epidemiology. This paper (https://doi.org/10.3390%2Fijms22189971) can meet these needs. A lecture is suggested.

Response: Thank you for your valuable suggestion. We have revised the introduction section to include more specific data regarding the annual incidence of PC and PC-related deaths. (page 2 line 48-50) We have also revised the reference #1 and #2 accordingly. (page 9 line 390-394)

Ref> 1. Siegel, R.L.; Miller, K.D.; Fuchs, H.E.; Jemal, A. Cancer Statistics, 2021. CA Cancer J. Clin. 2021, 71, 7–33. doi: 10.3322/caac.21654. Epub 2021 Jan 12

  1. Ferro M, de Cobelli O, Vartolomei MD, Lucarelli G, Crocetto F, Barone B, Sciarra A, Del Giudice F, Muto M, Maggi M, Carrieri G, Busetto GM, Falagario U, Terracciano D, Cormio L, Musi G, Tataru OS. Prostate Cancer Radiogenomics-From Imaging to Molecular Characterization. Int J Mol Sci. 2021 Sep 15;22(18):9971. doi.org/10.3390%2Fijms22189971

Reviewer 2 Report

The present study evaluates the implications of androgen deprivation therapy on the development of osteoporotic fractures in patients with prostate cancer. The topic is relevant, but there is a need for major changes to improve the initial form:

Shape suggestion

Abbreviations are explained when they first appear in the abstract or main text and contribute to making the text easier to read and the information conveyed more efficiently. Once an abbreviation has been established and explained, it will be used throughout the entire manuscript, with the exception of the abstract, where it must be treated separately. Please revise the whole manuscript and explain the abbreviations used directly, without explanation (e.g., L34- OS etc.).

The presence of the bibliographic index [134] in L66 is not in accordance with the rules adopted by the journal. Bibliographic indices will appear in the text in the order of their insertion. Please review the entire manuscript and proofread according to the instructions for authors.

L246-271 The information is organized in the form of an overly long paragraph, which decreases readability and comprehension. Please reorganize into shorter paragraphs that will be more logical and easier to understand.

Content suggestions

The aim of the paper presented in the last paragraph of the introduction needs to be improved from the perspective of describing the contribution to the field under analysis and the elements of scientific novelty presented.

It is advisable to discuss in more detail the defining elements of prostate cancer, along with the different therapeutic targets in this pathology. I suggest checking and referring to: PMID: 36677808.

It is necessary to explain in detail what ADT means, and which molecules act pharmacologically in this situation (use of LHRH and GnRH antagonists etc.). Furthermore, in the current context, a focus on the safety profile is also necessary.

Because of the various adverse effects of therapies used in cancer, it is advisable to present which adjuvant therapies based on biocompounds may have beneficial effects on the overall management. I suggest checking and referring to: PMID: 32405942.

The present study has multiple limitations due to the numerous parameters not taken into account, which may have a major influence on the final outcomes. In this regard, it is mandatory to emphasize the novelty of this study, not being the only one in this research direction, and to present serious future research directions to solve the limitations and present the future of research in this field.

Author Response

  • • Manuscript ID: cancers-2272182
  • Title: Association of androgen deprivation therapy with osteoporotic fracture

in patients with localized and regional prostate cancer using a retrospective

population-based propensity score-matched cohort

Reply to Reviewer #2’s comments

We thank reviewer #2 for their time spent reviewing our manuscript and valuable comments.

Shape suggestion

  1. Abbreviations are explained when they first appear in the abstract or main text and contribute to making the text easier to read and the information conveyed more efficiently. Once an abbreviation has been established and explained, it will be used throughout the entire manuscript, with the exception of the abstract, where it must be treated separately. Please revise the whole manuscript and explain the abbreviations used directly, without explanation (e.g., L34- OS etc.).

Response: Thank you for pointing this out. We have revised the manuscript accordingly. (page 1 line 32,34; page 4 line 157 and 179; page 7 line 313 and 327)

  1. The presence of the bibliographic index [134] in L66 is not in accordance with the rules adopted by the journal. Bibliographic indices will appear in the text in the order of their insertion. Please review the entire manuscript and proofread according to the instructions for authors.

Response: We corrected the reference number (page 2 line 83)

  1. L246-271 The information is organized in the form of an overly long paragraph, which decreases readability and comprehension. Please reorganize into shorter paragraphs that will be more logical and easier to understand.

Response: Thank you for your constructive suggestion. We have revised the aforementioned paragraph into shorter paragraphs and sentences to improve the readability (page 6 line 250-279)

Content suggestions

  1. The aim of the paper presented in the last paragraph of the introduction needs to be improved from the perspective of describing the contribution to the field under analysis and the elements of scientific novelty presented.

Response: Thank you for pointing this out. We have revised the paragraph of the introduction section to clarify the novelty and differentiate our objective from those of previous studies. The novelty of this study is that the localized and regional SEER staged PC cohort was defined to analyze the association of the ADT with OF. Most of the previously published research papers focused on the PC wither without tumor staging or with the inclusion of advanced and metastatic PC. (page 2 lines 86- page 3 line97)

  1. It is advisable to discuss in more detail the defining elements of prostate cancer, along with the different therapeutic targets in this pathology. I suggest checking and referring to: PMID: 36677808.

Response: Thank you for recommending this reference. We have added some explanation to the discussion section about the effect of primary and secondary hormonal agents and their potential therapeutic mechanism and cited the aforementioned reference accordingly. (page 7 line 328 and page 10 line 465)

Ref: 29. Sharma A, Sharma L, Nandy SK, Payal N, Yadav S, Vargas-De-La-Cruz C, Anwer MK, Khan H, Behl T, Bungau SG. Molecular Aspects and Therapeutic Implications of Herbal Compounds Targeting Different Types of Cancer. Molecules. 2023 Jan 11;28(2):750 doi: 10.3390/molecules28020750.

  1. It is necessary to explain in detail what ADT means, and which molecules act pharmacologically in this situation (use of LHRH and GnRH antagonists etc.). Furthermore, in the current context, a focus on the safety profile is also necessary.

Because of the various adverse effects of therapies used in cancer, it is advisable to present which adjuvant therapies based on biocompounds may have beneficial effects on the overall management. I suggest checking and referring to: PMID: 32405942.

Response: Thank you for this valuable suggestion. We have added some information to the introduction section to explain the details of ADT and its major side effects. We have also provided a new reference accordingly. (page 2 line 44-48 and line 54-59)

Ref> 5.Sciarra A, Busetto GM, Salciccia S, Del Giudice F, Maggi M, Crocetto F, Ferro M, De Berardinis E, Scarpa RM, Porpiglia F, Carmignani L, Damiano R, Artibani W, Carrieri G. Does Exist a Differential Impact of Degarelix Versus LHRH Agonists on Cardiovascular Safety? Evidences From Randomized and Real-World Studies. Front Endocrinol (Lausanne). 2021 Jun 14;12:695170. doi.org/10.3389/fendo.2021.695170

  1. The present study has multiple limitations due to the numerous parameters not taken into account, which may have a major influence on the final outcomes. In this regard, it is mandatory to emphasize the novelty of this study, not being the only one in this research direction, and to present serious future research directions to solve the limitations and present the future of research in this field.

Response: Thank you for this comment. We have added sentences in the limitation section to emphasize the significance of our findings and our use of a PC cohort with low-tumor burden with longterm ADT therapy (page 8 line 351-355)

Reviewer 3 Report

In this study, the authors investigated the incidence and effects on survival of osteoporotic fracture (OF) in Korean patients with androgen-dependent prostate cancer (PC), especially in localized or regional Surveillance, Epidemiology, and End Results (SEER)-staged patients with or without androgen deprivation therapy. The authors found that ADT significantly increased the incidence of OF in the spine, hip, and wrist; however, those with OF did not show any significant differences in overall survival between the ADT and non-ADT groups.

Comments:

The reviewer has some concerns as follows:

1. There are several recent studies have addressed similar topics to this study, such as Kim et al., 2021, https://doi.org/10.1038/s41598-021-89589-3; Chen et al., 2023, https://doi.org/10.1371/journal.pone.0279981; van Oostwaard et al., 2023, https://doi.org/10.1016/j.jbo.2022.100465. Therefore, the novelty of this study is really not high.

2. The data of formal Tables or Figures within this manuscript are lacking. It is difficult to judge the validity of this study due to the lack of data.

3. In general, the presented results cannot support the conclusions.

Author Response

  • • Manuscript ID: cancers-2272182
  • Title: Association of androgen deprivation therapy with osteoporotic fracture

in patients with localized and regional prostate cancer using a retrospective

population-based propensity score-matched cohort

Reply to Reviewer #3’s comments

We thank reviewer #3 for their time spent reviewing our manuscript and valuable comments.

There are several recent studies have addressed similar topics to this study, such as the following three articles. Therefore, the novelty of this study is really not high.

  1.  

Response:

Thank you for recommending the three articles which are important in the subject of osteoporotic fracture and the ADT. We have reviewd all three articles and found that there are big difference in the cohort selection and methodology. We have noted some of these differences between these studies and our study. The main difference is that our study was aimed to focus on the localized and regional SEER staged PC that indirectly reflected low tumor burden of PC with a possibility of long-term ADT due to low cancer-specific death rate. However, the first two articles did not satisfy patients based on the tumor staging system and most patients had advanced and metastatic PC with higher tumor burden and poorer performance states with shorter ADT duration. The third article used the TNM staging for their analytic purpose but a great portion of their cohort also consisted of patients with advanced PC and the sample size was also smaller compared with our study. Additionally, we aimed to define the relationship between OF and ADT in PC patients with low staged tumor burden with different OF sites and evaluate the effect of OF on overall survival, as well as identify the risk factors of OF. We have added this novelty at the end of the introduction section (page 2 line 86-97) and at the end of the discussion section. (page 6 line 249-252, page 8 line 349-252, line 354-360) We have also added two new references  accordingly (page 10 lines 447-452 and line 466-468)

Ref 22.Kim DK, Lee HS, Park JY, Kim JW, Ahn HK, Ha JS, Cho KS. Androgen-deprivation therapy and the risk of newly developed fractures in patients with prostate cancer: a nationwide cohort study in Korea. Sci Rep. 2021 May 12;11(1):10057. doi: 10.1038/s41598-021-89589-3.

  • This study did not stratify PC patients by the stage and most patients after propensity score matching had more advanced and metastatic disease. Patients treated with ADT for more than 3 years comprised only 25.5% of the cohort, linking this portion with low tumor burden. This implied that patients with more advanced and metastatic tumor burden and poorer performance states were included in the study, resulting in higher rate of osteoporotic fracture (page 6 line 250-279)

Ref 23.Chen WC, Li JR, Wang SS, Chen CS, Cheng CL, Hung SC, Lin CH, Chiu KY, Liao PC. Conventional androgen deprivation therapy is associated with an increased risk of fracture in advanced prostate cancer, a nationwide population-based study. PLoS One. 2023 Jan 4;18(1):e0279981. doi: 10.1371/journal.pone.0279981. eCollection 2023

  • This study also focused on the relationship between ADT and osteoporotic fracture without any stratification of the tumor burden. No clinical stage was implicated in the cohort selection and patients with more advanced disease were involved in the analysis.

Ref 30. van Oostwaard MM, van den Bergh JP, van de Wouw Y, Janssen-Heijnen M, de Jong M, Wyers CE. High prevalence of vertebral fractures at initiation of androgen deprivation therapy for prostate cancer. J Bone Oncol. 2022 Dec 7;38:100465. doi: 10.1016/j.jbo.2022.100465

  • This study stratified the PC cohort by risk criteria based on the stage. However, most of the included patients had advanced PC and many neoadjuvant ADTs were included in the study. This suggested that locally advanced PC patients whose tumor burden was quite high despite no visible bone metastasis were included. After previously treated ADT and M1b patients, they included patients with local disease defined as the use of intermediate-risk neoadjuvant ADT prior to radiotherapy, use of high-risk adjuvant ADT and radiotherapy, or advanced PCa defined as long-term palliative ADT treatment due to non-skeletal metastasis or biochemical recurrence. Another limitation of the study is the small number of cases included. (page 8 line 334-357)

  1. The data of formal Tables or Figures within this manuscript are lacking. It is difficult to judge the validity of this study due to the lack of data.

Response: Thank you for this suggestion. We have revised the table and figures to better to convey our results.

  1. In general, the presented results cannot support the conclusions.

Response: Thank you for this comment. We have revised the results and conclusion sections based on our analytical results. (page 5 line 209-217, and page 8 lines 354-360)

Other changes

  1. We added a phrase in the title to specify the characteristics of tumor burden in the PC cohort of the study. (page 1 line 3)
  2. We edited some of the names of the variables used for the joint subtypes. (page 12, 14 and 17)

Round 2

Reviewer 1 Report

Authors answered all comments and suggestions.

Author Response

We thank reviewer #1 for their time spent reviewing our manuscript and for valuable comments.

Reviewer 2 Report

The authors responded to my requests,

Author Response

We thank reviewer #2 for their time spent reviewing our manuscript and valuable comments.

Reviewer 3 Report

Where are the Figures 1-3 shown in text? 

I still don't see the formal Figures 1-3 presented in the text.

Author Response

We thank reviewer #3 for their time spent reviewing our manuscript and valuable comments.

Comment 1. Where are Figures 1-3 shown in text? 

Response: We checked the original file with figures #1-#3 and the figures were attached to the original manuscript.

Round 3

Reviewer 3 Report

This revised manuscript can be accepted.

No further comments.

Author Response

We thank reviewer #3 for their time spent reviewing our manuscript and valuable comments.

All the figures were also submitted with the original manuscript.